# Putting situational affordances in an intervention context: How the interaction between personality and intervention situations can help us explain differential intervention responses

Esther C. A. Mertens[1¤a¤b], Isabel Thielmann[2], Annalaura Nocentini[3], Aniek M. Siezenga[1,2], Jean-Louis van Gelder[1,2] *

1 Institute of Education and Child Studies, Leiden University, Leiden, the Netherlands, 2 Department of Criminology, Max Planck Institute for the Study of Crime, Security and Law, Freiburg im Breisgau, Germany, 3 Department of Educational Sciences and Psychology, University of Florence, Firenze, Italy

¤a Current address: Department of Criminology, Max Planck Institute for the Study of Crime, Security and Law, Freiburg im Breisgau, Germany
¤b Current address: Netherlands Institute for the Study of Crime and Law Enforcement, Amsterdam, the Netherlands
* j.vangelder@csl.mpg.de

**Data Availability Statement:** Data supporting the findings of this study are openly available in the

## Abstract

We propose a framework in which interventions are described as situations affording the expression of certain personality traits to provide a systematic understanding of differential intervention response by personality traits. The goal of the present paper is twofold: 1) elaborate on the proposed framework, and 2) provide an initial test of this framework. We empirically tested this framework using data from a Randomized Controlled Trial ($N$ = 176) that examined a smartphone-based intervention aimed at increasing future-oriented thinking and behavior, and assessed HEXACO personality traits. The results showed that more introverted and agreeable individuals profited most from the intervention. Although these results were not in line with our a priori predictions, they could be explained using the proposed situational affordances framework. This shows the potential of this framework in an intervention context, though more research and tests using different interventions are needed.

**Trial registration:** The trial is registered in the Netherlands Trial Register number NL9671. Additionally, the hypotheses and analysis plan of the present study were pre-registered (AsPredicted #94684; https://aspredicted.org/95F_CDR).

## Introduction

Interventions play an important role in society and are implemented in a broad variety of domains (e.g., parenting interventions, educational interventions, health interventions).

Center for Open Science Online Supporting Information at https://osf.il/zxndf/.

**Funding:** This work was financially supported by a European Research Council Consolidator Grant (no. 772911-CRIMETIME). The funder had no role in study design, data collection and analysis, decision to publish, or preparation of the manuscript".

**Competing interests:** The authors have declared that no competing interests exist.

However, research consistently shows that the effectiveness of interventions differs between individuals, implying that some individuals profit more from certain interventions than others. Identifying individual characteristics related to intervention responsiveness is thus pivotal for optimizing intervention success. In the present study, we propose that interventions place individuals in different situations that may afford the expression of different personality traits and, thereby, allow different individuals to benefit more, or less, from that specific intervention. More specifically, we examined whether intervention situations created by the FutureU intervention moderated the relation between individuals' personality traits and the intervention's effects on future self-identification.

## Personality and intervention effects

Prior research has considered multiple individual-level moderators of intervention effects, including personality traits (e.g., [1, 2]). However, a systematic understanding about which personality traits moderate the effectiveness of which type of intervention remains elusive due to inconsistent moderation patterns across studies. For example, whereas Extraversion and Openness to Experience moderated intervention effects of a gratitude-based intervention [3], these traits did not moderate effects of a universal mindfulness intervention [4]. Likewise, whereas high levels of Conscientiousness were related to intervention resistance in an indicated school-based intervention [2], they were associated with stronger intervention effects in an indicated clinical intervention [5].

To address this issue, we propose a theoretical framework that intends to provide the basis for a systematic understanding of the dependency of intervention effects on personality traits. The aim of the present study is twofold. First, we describe interventions in terms of situations that allow for the expression of certain personality traits in behaviors. Dependent on the match between these situations and traits, individuals may benefit more, or less, from an intervention. Second, we take a first step towards testing this framework empirically using data form a study evaluating a specific intervention (i.e., FutureU) aimed at strengthening people's future self-identification.

## Situational affordances in an intervention context

Different explanations for why interventions may be more or less effective for different people have been proposed. One individual-level explanation suggests that certain personality traits can make individuals more susceptible to environmental influences (i.e., differential susceptibility hypothesis; [6]). For example, negative emotionality has been associated with deeper processing of environmental stimuli potentially making individuals more susceptible to their environment [6]. Given that an intervention can be considered an environmental factor, these personality traits may thus foster intervention effects [1, 7]. However, the differential susceptibility hypothesis suggests a consistent pattern of moderation by personality traits across different interventions and is therefore unable to explain differences in moderation patterns as typically observed in research. Another explanation focuses on the contextual level, suggesting that the context in which the intervention is embedded can support or obstruct individuals to use the learned intervention techniques [8, 9]. Although this explanation can account for differences in intervention effects between individuals in different contexts, it does not relate these differences to personality. Moreover, neither of the two explanations addresses characteristics of the intervention itself.

To explain differential responsiveness to interventions, we propose to consider interventions in terms of multiple situations, each of which provides specific affordances for the expression of certain personality traits. Situations have specific characteristics that enable, i.e.,

*afford*, a range of possible behaviors [10]. As such, situational affordances can activate, or inhibit, certain personality traits, which forms the basis for engaging in behaviors related to the activated trait. Affordances can thus provide a framework to organize and understand variation in individuals' behavioral responses to situational circumstances as a function of their personality [10–12]. For instance, situations that allow for social interactions are likely to activate the trait of Extraversion: Whereas extraverted individuals typically react with enthusiasm in social situations, introverted individuals may retract from such situations [13].

We propose that interventions can also be understood in terms of affordances. By participating in an intervention, individuals encounter situations related to the intervention's program and content, and to expectations related to participating in an intervention. For example, participants are expected to attend the intervention sessions and participate in the intervention's assignments and exercises. In other words, interventions put participants in 'intervention situations' that afford the expression of different personality traits, thus eliciting different responses by individuals as a function of their trait levels in which some situations of an intervention may 'fit' people and other situations may not. To illustrate, interventions using role-play create social interactions which may afford the expression of Extraversion, thereby potentially allowing extraverted individuals to benefit more from this assignment than introverted participants, as it 'fits' the former better; extraverted individuals fully engage in the role-play which allows them to learn from this assignment, while introverted individuals may not feel at ease and disengage from the role-play exercise which decreases their opportunity to learn from it. As such, an affordance-based framework also allows deriving predictions about which personality traits should moderate intervention effects for specific types of intervention.

## The affordance-based intervention framework: An empirical test using the FutureU intervention

As a first test of this affordance-based intervention framework, we used data from a study examining a smartphone-based intervention–FutureU–that aims to strengthen people's identification with their 'future self' in order to increase their future-oriented thinking and behavior. The intervention is based on the assumption that short-sighted decision making (versus future-oriented decision making) can be attributed to a lack of psychological connection between temporally distinct selves [14, 15]. Increasing the degree to which people identify with their future self is assumed to increase their tendency to make decisions that favor their future self over their present self [15]. Note 1) that FutureU does not aim to change personality traits, though response to this intervention may depend upon personality traits, and 2) that we examine one specific intervention as a first step towards empirically testing the herein proposed affordance-based framework.

The FutureU intervention entails various intervention situations. Two of these situations apply to almost all interventions: The intervention creates a situation in which participants 1) are asked to adhere to its sessions and 2) learn new skills. Specifically, FutureU is implemented via a smartphone application (app) that requires a daily check-in for three consecutive weeks. Interactions and exercises in the app are aimed at teaching new skills, such as thinking about future consequences when making a decision, and creating new (self-)insights, such as insight into their personality traits. Besides these common intervention situations, FutureU creates more specific types of situations that only apply to a subset of interventions: Goal-setting and social interaction. That is, participants are asked to set goals for themselves and work towards those goals. Moreover, participants interact with their future self–a digitally-aged rendering of themselves–thus providing situations that involve sociability. Even more specific to FutureU, the intervention involves interactions with a 'future self'-avatar and requests to make decisions

**Table 1. Overview of FutureU situations, affordances, and related personality traits.**

| Situation | Affordance for... | Personality trait |
|---|---|---|
| Treatment adherence by daily opening the FutureU app | Being planful and organized | Conscientiousness |
| Setting goals during the intervention | Showing goal-directed behaviors | Conscientiousness |
| Temporal conflict in which participants delay gratification favoring future gains over immediate gains | Being planful and self-control | Conscientiousness |
| New skills taught during the intervention | Being curious and eager to learn | Openness to Experience |
| (Self-)insights gained during the intervention | Favoring novel experiences and eager to learn | Openness to Experience |
| Future self, the idea and virtual rendering of who the participant may be in the future | Favoring novel, experimental, and unconventional experiences | Openness to Experience |
| Social interaction with the future self | Being talkative and enthusiastic | Extraversion |
| Valence towards the future self | Being enthusiastic and lively | Extraversion |

Participants in the control condition did not encounter these situations except for the situation "Setting goals during the intervention".

favoring this future self. These future self-interactions require participants to be open to the idea of a future self, accept the rendering of their future self, and experience a positive attitude towards their future self. As such, the intervention creates situations characterized by a temporal conflict between the needs and wants of the present self and those of the future self, asking participants to delay gratification by favoring the needs and wants of the future self over those of the present self. Except for goal-setting, these intervention situations were not apparent in the control condition.

These different intervention situations created by the intervention afford the expression of different personality traits as captured by structural models of personality, such as the HEXACO model [13, 16]. Specifically, situations characterized by treatment adherence (i.e., using the app daily), setting goals, and temporal conflict (i.e., delayed gratification) arguably afford the expression of Conscientiousness. People with high levels of Conscientiousness are drawn to organized situations and situations that require task- or goal-oriented behaviors. They are typically planful, organized, pursue their goals even under distracting circumstances, and demonstrate self-control. Furthermore, situations characterized by acquiring new skills and insights as well as interactions with the future self afford the expression of unconventional ideas and imagination which are embedded in Openness to Experience. People with high levels of Openness to Experience favor novel, experimental, and unconventional situations. They are curious, imaginative, and like to learn new things. Third, situations characterized by social interactions and valence towards others (e.g., the future self) afford the expression of Extraversion. People with high levels of Extraversion are likely to engage in social endeavors. They usually experience social situations as rewarding and are talkative, enthusiastic, lively, and socially bold. Taken together, the characteristics of the situations describing FutureU should mainly afford the expression of Conscientiousness, Openness to Experience, and Extraversion (Table 1).

## The present study

The present study aimed to illuminate the potential of situational affordances for improving our understanding of personality influences in intervention contexts. To this end, we described the FutureU intervention in terms of intervention situations and corresponding affordances and examined whether the intervention situation (compared to the control condition) moderated the relation between personality traits and intervention effects. The FutureU

intervention situations were not apparent in the control condition, except for setting goals. As in the intervention condition, people in the control condition set goals, creating a situation affording goal-directed behaviors, but in contrast to the intervention condition they received no further intervention and, thus, did not encounter other intervention situations.

We hypothesized that people with high scores on the personality traits afforded by FutureU intervention situations–that is, Conscientiousness, Openness to Experience, and/or Extraversion–would profit more from the intervention, thus showing stronger increases in their future self-identification than people scoring lower on these traits. In other words, we hypothesized that personality traits interact with the intervention situation in predicting intervention effects. We also explored potential interactions of the other three traits from the HEXACO personality model, i.e., Honesty-Humility, Emotionality, and Agreeableness, without specifying hypotheses. Furthermore, we expected a direct, positive effect of the FutureU intervention on future self-identification given that it aims to increase people's identification with their future self. We had no specific hypotheses regarding direct effects of personality traits on future self-identification.

## Method

### Design and procedure

The present study used data of a Randomized Controlled Trial (RCT) which consisted of an intervention condition and an active control condition (Fig 1). During the intake participants gave written informed consent, completed the baseline questionnaire, and set goals (see details below).

Data collection took place from 12 October 2021 to 21 March 2022. Questionnaires were completed at baseline (T1 –during the intake), after each week-long intervention module (T2 and T3), and immediately after the intervention (T4), or at parallel timepoints for the control condition. The RCT was approved by the Ethics Board of the Institute of Education and Child Studies at Leiden University (ECWP2021-320) and registered in the Netherlands Trial Register number NL9671 (see [17] for the study protocol). The current study was pre-registered specifying the hypotheses and analysis plan (AsPredicted #94684; https://aspredicted.org/95F_CDR).

### Participants

In total, 176 first-year university students participated in the study ($n_{intervention}$ = 87, $n_{control}$ = 89; $M_{age}(SD)$ = 19.5(2.8); 88% women) of whom 101 (57%) studied pedagogical sciences and 66 (38%) psychology. There were no reliable differences in participants' age between conditions (Intervention $M_{age}$ = 19.7; Control $M_{age}$ = 19.2), but there were slightly less women in the intervention condition (82%) than in the control condition (94%).

During data collection there was a COVID-19 related lockdown. Generally, there were no differences between participants included before and after the lockdown (see supporting information Participants).

### Missing data

Data were regarded as missing when participants did not complete a questionnaire at all or in time (i.e., within four days on T2 and T3, and within eight days on T4). At T2, T3, and T4, respectively 4, 5, and 6 participants did not complete the questionnaire in time. Across all timepoints, 2.16% of the data were missing. Missing data were missing at random (Little's MCAR test: $\chi^2$ (103) = 128.88, $p$ = .043).

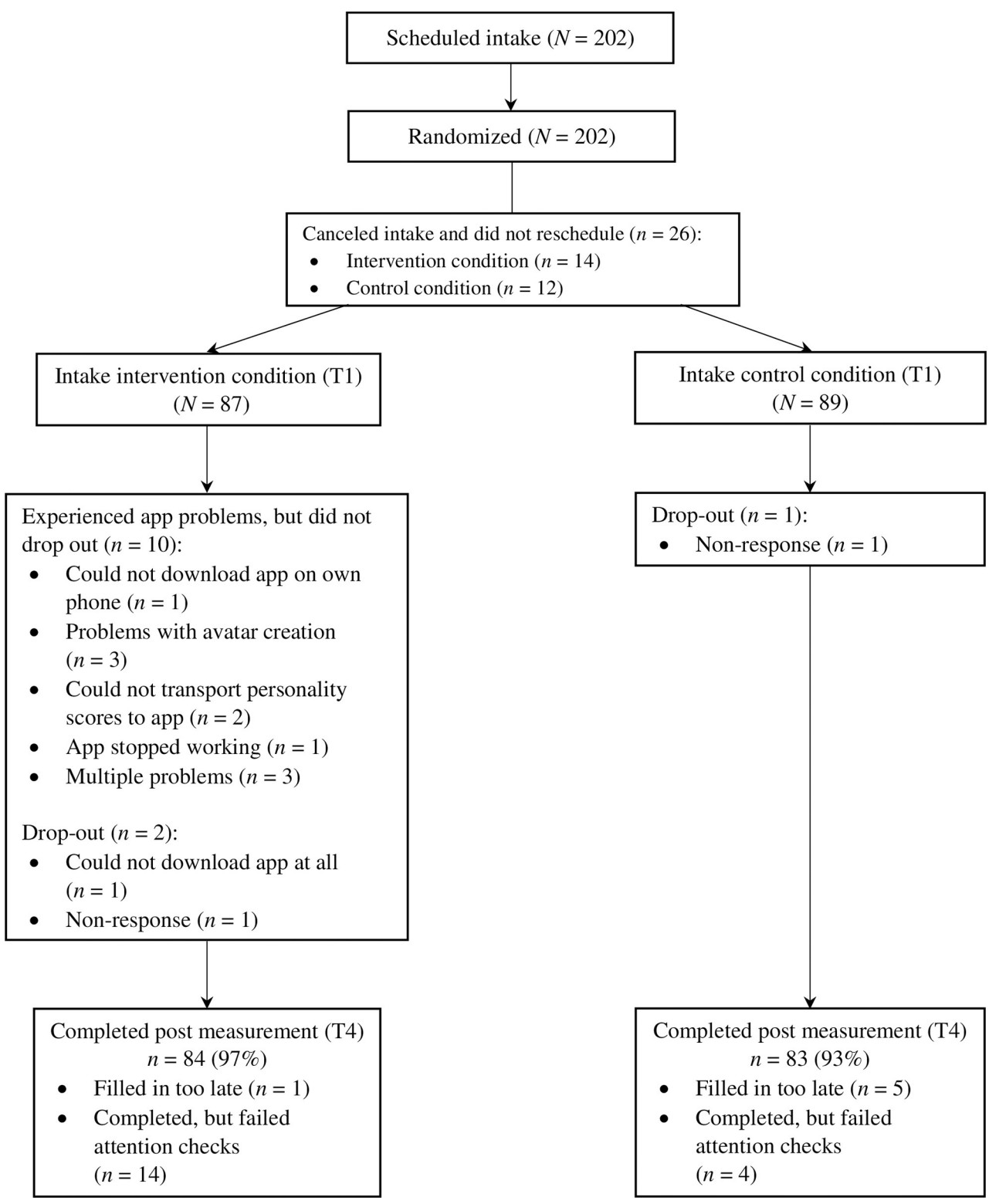

**Fig 1. Flow chart.**

## Conditions

**Intervention condition: FutureU.**   During the intake, participants set a goal they wanted to achieve within a year and a goal that they wanted to achieve within a month. Subsequently, both at intake and during the intervention, they set weekly goals as intermediate steps towards their monthly goal. Given that goal commitment and achievement are higher when goals are set by individuals themselves [18], there were no restrictions regarding the type of goals participants could set. At the end of the intake, participants took a photo of their face (i.e., a 'selfie'), which was used to create a digitally aged avatar representing their future self, and subsequently downloaded the FutureU app on their smartphone.

Participants received daily push notifications from the app. To open the app, participants 'connected' with their future self by touching the (virtual) finger of the future self on the blurred screen after which the screen unblurred and the avatar became visible. After this 'connection mechanic', participants were directed to the chat menu where they 'interacted' with their future self via scripted messages. In the chat, participants received psychoeducation, responded to questions about their future to motivate them to contemplate the future, and received instructions for assignments. Other app features progressively unlocked during the intervention, namely, a personal profile of the future self which was completed by the participant, an overview of personality scores of both present and future self (personality scores of the present self were derived from the personality questionnaire completed by the participant at baseline, personality scores of the future self were set by the participant), a time travel portal in which participants switch between the perspectives of the present and future selves, and a scheme in which participants can fill in their goal, potential obstacles, and a plan to overcome these (Fig 2, for more information see [17]).

*Treatment adherence*. The app was designed for daily use, i.e., 21 days, for approximately 5 minutes or less. Participants used the FutureU app on average for 14 days (*SD* = 4.73; Median = 15 days) for 4.39 minutes (*SD* = 1.71) per day. Eight (9.20%) participants checked-in

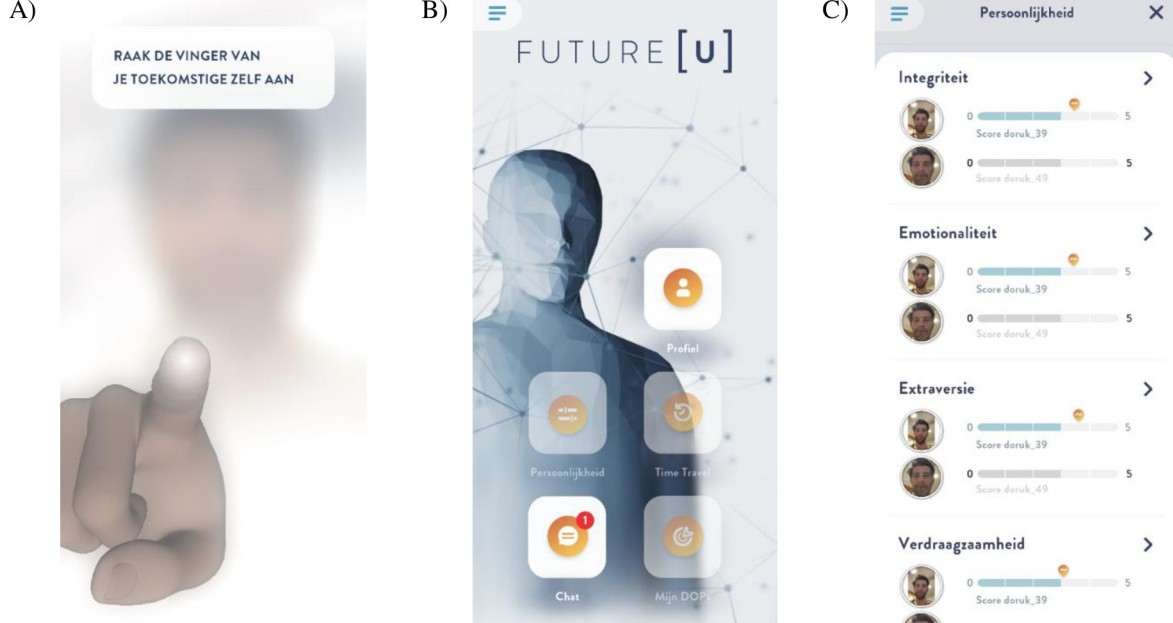

**Fig 2. Screenshots of the FutureU app.** A) the connection mechanic, B) the home screen, and C) the personality profile. Printed with permission from the copyright holder.

every day and 32 (36.78%) participants checked-in frequently (check-in range = 16–20 days). Some participants (*n* = 10, 11.49%) experienced technical problems with the app and one (1.15%) participant was not able to download the app and dropped out (Fig 1).

**Control condition.** Via the same procedure as in the intervention condition, participants in the control condition set two goals during the intake. Subsequently, participants independently set weekly goals at the intake and during the next three weeks. Contrary to the intervention condition, participants received no further support to achieve their goals. Hence, besides setting weekly goals for three weeks, the situations encountered by the participants in their daily lives were not affected by an intervention.

## Measurements

**Intervention outcome: Future self-identification.** Future self-identification (the intervention's key outcome) was measured at T1 through T4 with three indices, i.e., vividness of, valence towards, and relatedness to the future self.

*Vividness.* The degree to which people have a vivid and clear image of their future self was assessed with five items based on Van Gelder et al. [19]. Items (e.g., "I have a clear image of myself in 10 years.") were answered on a 7-point rating scale (1 = *completely disagree* to 7 = *completely agree*; T1 –T4 α = .92 - .95).

*Valence.* Positive feelings towards the future self were assessed with one item based on Hershfield et al. [20]: "How do you feel when you think about your future?". The item was answered via the 9-point Self-Assessment Manikin ranging from negative feelings to positive feelings [21], with higher scores representing more positive feelings.

*Relatedness.* The extent to which people feel connected and similar to their future self was assessed with the two-item Future Self-Continuity Measure [20]: "How connected do you feel to your future self?" and "How similar do you feel to your future self?". Items were answered on a 7-point scale on which each point is marked by a pair of circles that increase in overlap across the scale–more overlap represents more connectedness or similarity, respectively, with the future self. Reliability was generally good (T2 –T4 α = .77 - .83), though relatively low at T1 (α = .57).

**Moderators: Personality traits.** Personality was assessed at baseline using the HEXACO-100 [22] measuring the six HEXACO personality traits: Honesty-Humility (e.g., "I am an ordinary person who is no better than others.", α = .84), Emotionality (e.g., "I sometimes can't help worrying about little things.", α = .82), Extraversion (e.g., "In social situations, I'm usually the one who makes the first move.", α = .86), Agreeableness (e.g., "I generally accept people's faults without complaining about them.", α = .84), Conscientiousness (e.g., "When working, I often set ambitious goals for myself.", α = .87), and Openness to Experience (e.g., "I would enjoy creating a work of art, such as a novel, a song, or a painting.", α = .82). Each trait is assessed with 16 items answered on a 5-point rating scale (1 = *completely disagree* to 5 = *completely agree*).

## Analyses

We first conducted preliminary analyses to test intervention effects across time for each of the three future self-identification outcome measures (i.e., vividness, valence, and relatedness) in SPSS (version 27) using linear mixed regression models. Maximum likelihood was used to handle missing data. We dummy-coded condition with the control condition as the reference group. In the models, this dummy variable of condition was added as a predictor together with a two-way interaction of time * condition. A dummy variable representing gender (0 = male, 1 = female) was added as a covariate, given that the conditions differed on gender at baseline.

Interactions of personality traits with intervention situations on the three future self-identification outcome measures (i.e., vividness, valence, and relatedness) were also analyzed with

linear mixed regression models for each outcome separately. Condition was represented with a dummy variable and all trait variables were mean-centered. To test the moderation effect regarding personality traits in the intervention condition across time, we added a three-way interaction: Time * condition * personality trait. (Models including all two-way interactions related to the three-way interactions showed roughly the same results and are reported in the supporting information Linear mixed models and S2 and S3 Tables in S1 File) The dummy variable of gender was added as a covariate and missing data was handled with maximum likelihood estimation.

We used slightly different models for examining moderation concerning the three traits of main interest (the primary analyses) and the other three traits (the secondary analyses). In our primary analyses, we examined the associations of Conscientiousness, Openness to Experience, and Extraversion (the traits of main interest) with future self-identification. Condition and the three trait variables were included as predictors of the respective outcome in the model, both as main effects and as interaction effects. In our secondary analyses, we explored associations of Honesty-Humility, Emotionality, and Agreeableness with future self-identification. In these models, we included condition and the six trait variables as predictors of the respective outcome in the model (i.e., main effects). In addition, we added one interaction variable at a time as predictor of the outcome (i.e., interaction-effect), resulting in three models for each outcome.

For significant interaction effects, separate effect sizes were calculated for subgroups scoring low, average, and high ($M \pm 1$ $SD$) on the relevant trait. We calculated effect sizes per intervention week (i.e., change from T1 to T2, from T2 to T3, and from T3 to T4) and the overall effect (i.e., change from T1 to T4) using the following formula [23]:

$$\text{Cohen's } d = \frac{M \text{ change intervention}}{SD \text{ pooled}} - \frac{M \text{ change control}}{SD \text{ pooled}}$$

The robustness of our results was examined using sensitivity analyses. To this end, two attention check items were embedded in the T4 questionnaire with the instruction to mark a specific response category as local indicators of participants' (in)attentiveness [24]. We excluded participants who failed one or both attention checks or who experienced technical issues with the app ($n = 32$). Subsequently, we reran the models described above with this reduced sample ($N = 144$).

Codes used for the analyses are openly available in the Center for Open Science Online Supporting Information at https://osf.io/zxndf/

## Results

### Preliminary analyses

Descriptive statistics and correlations of the future self-identification outcomes per assessment and of personality traits are presented for the complete sample in Table 2 (descriptives per condition are presented in S1 Table in S1 File). The linear mixed regression models to examine the intervention's effectiveness on future self-identification showed a small intervention effect over time on vividness ($F(3, 303.50) = 2.88$, $p = .036$, $d_{\text{Overall}} = .23$ 95%CI $d_{Overall} = -.07$; .52), which seemed to have emerged especially during the first week of the intervention (i.e., from T1 to T2; $d_{\text{T1-T2}} = .22$). There were no intervention effects for valence ($F(3, 504.39) = 0.04$, $p = .988$, $d_{\text{Overall}} = .00$ 95%CI $d_{Overall} = -.30$; .30) nor for relatedness ($F(3, 503.57) = 0.43$, $p = .429$, $d_{\text{Overall}} = .13$ 95%CI $d_{Overall} = -.17$; .43).

**Table 2. Descriptive statistics and correlations of the outcome variables at each measurement occasion and of personality traits of the total sample (N = 176).**

| | M | SD | 1 | 2 | 3 | 4 | 5 | 6 | 7 | 8 | 9 | 10 | 11 | 12 | 13 | 14 | 15 | 16 | 17 |
|---|---|---|---|---|---|---|---|---|---|---|---|---|---|---|---|---|---|---|---|
| **1 Vividness T1** | 3.37 | 1.46 | - | | | | | | | | | | | | | | | | |
| **2 Valence T1** | 6.66 | 1.39 | .44** | - | | | | | | | | | | | | | | | |
| **3 Relatedness T1** | 3.85 | 1.07 | .36** | .45** | - | | | | | | | | | | | | | | |
| **4 Vividness T2** | 3.80 | 1.32 | .76** | .43** | .38** | - | | | | | | | | | | | | | |
| **5 Valence T2** | 6.55 | 1.28 | .35** | .70** | .36** | .42** | - | | | | | | | | | | | | |
| **6 Relatedness T2** | 3.89 | 1.09 | .31** | .36** | .70** | .38** | .39** | - | | | | | | | | | | | |
| **7 Vividness T3** | 3.89 | 1.40 | .72** | .43** | .40** | .87** | .42** | .43** | - | | | | | | | | | | |
| **8 Valence T3** | 6.45 | 1.22 | .35** | .60** | .40** | .41** | .77** | .42** | .46** | - | | | | | | | | | |
| **9 Relatedness T3** | 4.06 | 1.03 | .26** | .31** | .62** | .35** | .38** | .80** | .45** | .47** | - | | | | | | | | |
| **10 Vividness T4** | 3.80 | 1.40 | .76** | .49** | .40** | .84** | .50** | .42** | .86** | .49** | .45** | - | | | | | | | |
| **11 Valence T4** | 6.52 | 1.27 | .33** | .58** | .31** | .37** | .70** | .43** | .42** | .67** | .35** | .50** | - | | | | | | |
| **12 Relatedness T4** | 4.18 | 1.08 | .24** | .31** | .61** | .29** | .37** | .80** | .40** | .39** | .82** | .41** | .41** | - | | | | | |
| **13 Conscientiousness** | 3.51 | 0.62 | .21** | .14 | .21** | .19* | .09 | .12 | .13 | .07 | .13 | .17* | .22** | .14 | - | | | | |
| **14 Openness to Experiences** | 3.16 | 0.61 | -.01 | .02 | -.02 | -.03 | .01 | -.01 | -.03 | .04 | .01 | .02 | .04 | .07 | -.00 | - | | | |
| **15 Extraversion** | 3.47 | 0.55 | .18* | .47** | .22** | .20** | .41** | .28** | .24** | .38** | .23** | .27** | .34** | .28** | -.05 | -.04 | - | | |
| **16 Honesty-Humility** | 3.60 | 0.55 | -.10 | -.01 | .06 | -.02 | .01 | .02 | -.07 | -.06 | .08 | -.05 | .01 | .03 | .23** | -.02 | -.11 | - | |
| **17 Emotionality** | 3.41 | 0.56 | -.03 | -.09 | -.25** | -.11 | -.06 | -.10 | -.10 | -.03 | -.10 | -.10 | .03 | -.11 | .16* | -.03 | .00 | .02 | - |
| **18 Agreeableness** | 3.12 | 0.58 | -.11 | .05 | .02 | -.03 | .06 | -.00 | -.09 | .01 | .04 | .03 | .05 | .04 | .01 | .00 | .11 | .29** | -.00 |

T1 = Baseline; T2 and T3 = Interim measurements; T4 = Post measurement.

\* $p < .05$

\*\* $p < .01$

## Primary analyses: Conscientiousness, Openness to Experience, and Extraversion

Conscientiousness and Extraversion showed a positive main effect on all three outcomes, that is, on vividness of, valence towards, and relatedness to the future self. Higher scores on these traits were related to higher levels of vividness ($\beta_{Cns}(SE) = .33(.14)$; $\beta_{Ext}(SE) = .42(.14)$), valence ($\beta_{Cns}(SE) = .25(.12)$; $\beta_{Ext}(SE) = .55(.12)$) and relatedness ($\beta_{Cns}(SE) = .22(.11)$; $\beta_{Ext}(SE) = .33(.11)$). There were no significant main effects for Openness to Experience and condition.

Regarding the interaction effects, only one effect was statistically significant (Table 3): Intervention situations moderated the relation between Extraversion and intervention effects for valence towards the future self. Particularly in the first week of the intervention, in contrast to our hypotheses, lower levels of Extraversion were related to stronger intervention effects on valence ($d_{T2} = .52$; $d_{T3} = -.22$; $d_{T4} = -.04$; $d_{Overall} = .28$ 95%CI $d_{Overall} = -.02$; .57), whereas higher levels of Extraversion were related to weaker intervention effects on this outcome ($d_{T2} = -.53$; $d_{T3} = .20$; $d_{T4} = .15$; $d_{Overall} = .08$ 95%CI $d_{Overall} = -.22$; .38; Fig 3).

**Sensitivity analyses.** The results of the sensitivity analyses (reduced $N = 144$) were in line with the results of the analyses on the total sample (Table 3).

## Secondary analyses: Honesty-Humility, Emotionality, and Agreeableness

There was a negative main effect of Emotionality on relatedness to the future self, meaning that higher levels of Emotionality predicted lower levels of relatedness ($\beta_{Emo}(SE) = -.18(.07)$). As in the models of the primary analyses, we found positive main effects of Conscientiousness and Extraversion on all three outcomes (i.e., vividness, valence, and relatedness) with higher levels of these two traits predicting higher levels of vividness ($\beta_{Cns}(SE) = .31(.10)$; $\beta_{Ext}(SE) =$

**Table 3. Results of the linear mixed models regarding Conscientiousness, Openness to Experience, and Extraversion per outcome.**

| | Total sample (N = 176) | | | | | | Sensitivity analyses (N = 144) | | | | | |
|---|---|---|---|---|---|---|---|---|---|---|---|---|
| | Vividness | | Valence | | Relatedness | | Vividness | | Valence | | Relatedness | |
| | F | p | F | p | F | p | F | p | F | p | F | p |
| Time | 22.30** | < .001 | 2.04 | .107 | 12.55** | < .001 | 18.50** | < .001 | 2.07 | .104 | 10.70** | < .001 |
| Gender | 3.01 | .085 | 0.61 | .435 | 1.13 | .289 | 2.06 | .153 | 1.54 | .217 | 0.10 | .750 |
| Condition | 0.00 | .991 | 0.58 | .446 | 0.31 | .579 | 0.01 | .913 | 0.13 | .714 | 0.02 | .900 |
| Conscientiousness | 8.59** | .004 | 5.72* | .018 | 6.62* | .011 | 9.07** | .003 | 3.45 | .065 | 6.95** | .009 |
| Openness to Experience | 0.32 | .573 | 0.29 | .594 | 0.01 | .929 | 0.07 | .788 | 0.27 | .603 | 0.11 | .738 |
| Extraversion | 11.48** | .001 | 51.65** | < .001 | 15.80** | < .001 | 4.78* | .030 | 35.02** | < .001 | 7.64** | .006 |
| Time*Condition*Conscientiousness | 0.68 | .689 | 1.08 | .375 | 0.52 | .819 | 0.48 | .847 | 0.67 | .699 | 0.99 | .441 |
| Time*Condition*Openness to Experience | 1.14 | .335 | 0.77 | .609 | 1.05 | .395 | 0.51 | .825 | 0.34 | .937 | 0.71 | .664 |
| Time*Condition*Extraversion | 0.81 | .580 | 2.54* | .014 | 0.78 | .601 | 0.78 | .605 | 2.82** | .007 | 1.79 | .088 |

* p < .05

** p < .01

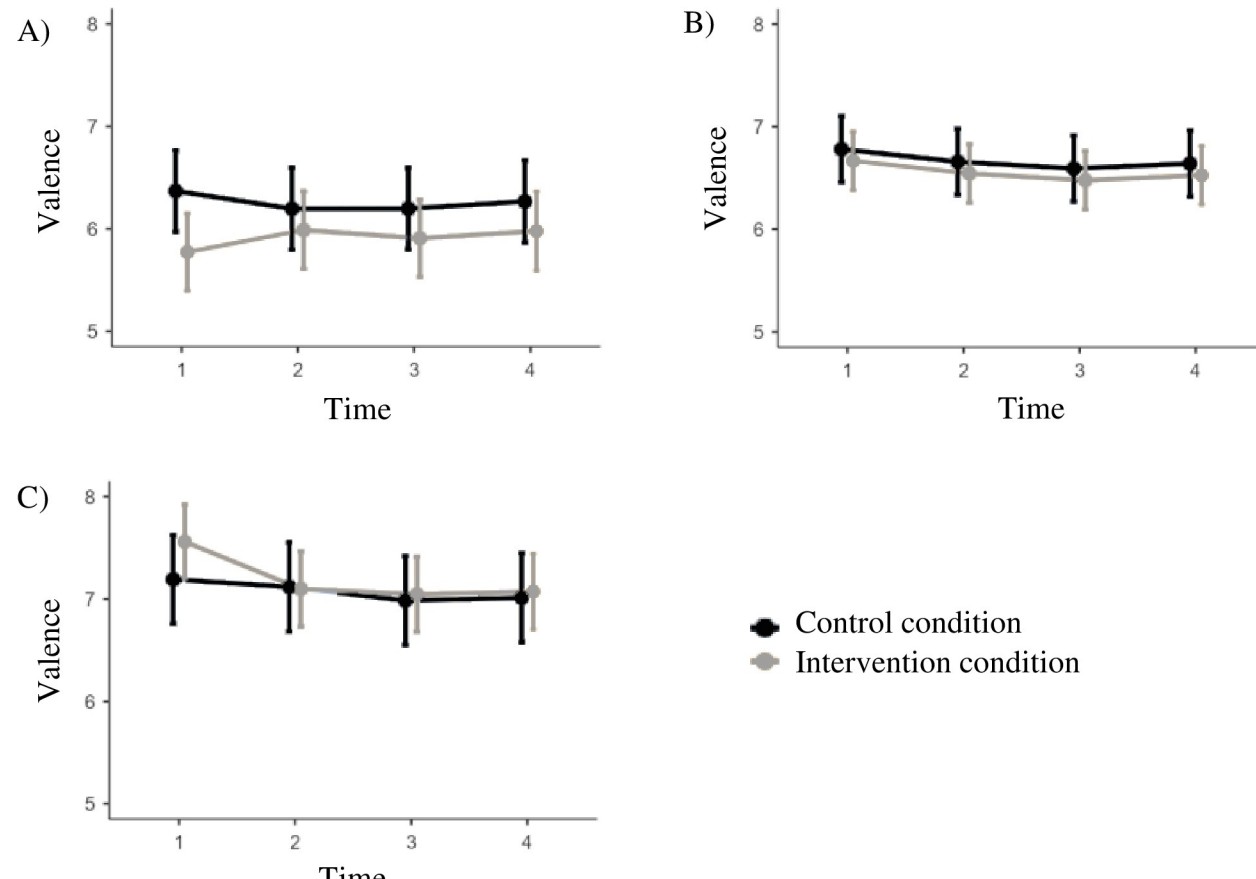

**Fig 3. Moderation of intervention effects on valence by Extraversion with plots of the intervention effects for participants.** A) low levels (M -1SD), B) average levels (M +/-1SD), and C) high levels (M +1SD) of Extraversion. The range of the y-axis is truncated.

**Table 4. Results of the linear mixed models regarding Honesty-Humility, Emotionality, and Agreeableness per outcome.**

| | Total sample ($N = 176$) | | | | | | Sensitivity analyses ($N = 144$) | | | | | |
|---|---|---|---|---|---|---|---|---|---|---|---|---|
| | Vividness | | Valence | | Relatedness | | Vividness | | Valence | | Relatedness | |
| | *F* | *p* | *F* | *p* | *F* | *p* | *F* | *p* | *F* | *p* | *F* | *p* |
| **Time** | 22.98** | < .001 | 1.71 | .163 | 12.74** | < .001 | 17.98** | < .001 | 1.56 | .198 | 11.50** | < .001 |
| **Gender** | 2.15 | .144 | 1.30 | .255 | 0.50 | .482 | 1.39 | .240 | 2.32 | .130 | 0.00 | .981 |
| **Condition** | 0.27 | .603 | 0.92 | .338 | 0.86 | .356 | 0.54 | .464 | 0.31 | .580 | 0.34 | .559 |
| **Conscientiousness** | 10.52** | .001 | 6.22* | .014 | 7.55** | .007 | 11.53** | .001 | 3.75 | .055 | 7.90** | .006 |
| **Openness to Experience** | 0.29 | .592 | 0.13 | .722 | 0.01 | .907 | 0.12 | .731 | 0.44 | .511 | 0.08 | .774 |
| **Extraversion** | 12.07** | .001 | 53.19** | < .001 | 17.94** | < .001 | 5.91* | .016 | 34.95** | < .001 | 9.81** | .002 |
| **Honesty-Humility** | 0.16 | .693 | 0.43 | .513 | 1.23 | .269 | 0.61 | .435 | 0.08 | .779 | 0.34 | .562 |
| **Emotionality** | 1.80 | .182 | 0.53 | .469 | 6.40* | .012 | 3.51 | .063 | 0.39 | .531 | 6.28* | .013 |
| **Agreeableness** | 1.59 | .209 | 0.42 | .518 | 0.70 | .404 | 1.19 | .278 | 0.02 | .898 | 0.89 | .347 |
| **Time*Condition*Honesty-Humility** | 0.56 | .792 | 0.68 | .687 | 0.48 | .847 | 0.50 | .838 | 1.05 | .395 | 0.72 | .659 |
| **Time*Condition*Emotionality** | 1.27 | .264 | 1.30 | .251 | 1.14 | .336 | 1.17 | .321 | 0.68 | .689 | 1.39 | .208 |
| **Time*Condition*Agreeableness** | 1.98 | .057 | 0.79 | .597 | 0.62 | .740 | 2.91** | .006 | 0.89 | .515 | 0.77 | .611 |

Reported statistics of main effects and covariates are based on the model that included the interaction-effect with Honest-Humility as these effects were highly similar across models.

* $p < .05$

** $p < .01$

.32(.09)), valence ($\beta_{Cns}$(*SE*) = .19(.08); $\beta_{Ext}$(*SE*) = .54(.07)) and relatedness ($\beta_{Cns}$(*SE*) = .19(.07); $\beta_{Ext}$(*SE*) = .28(.07)). There were neither significant main effects of the other personality traits nor of condition. There were also no significant interaction effects of any of the secondary traits with condition on the outcomes (Table 4).

**Sensitivity analyses.** In general, the findings of the sensitivity analyses replicated the results of the models conducted on the whole sample. However, the interaction effect of Agreeableness on vividness became significant (in the main analyses, there was only a trend; Table 4). Specifically, participants with high levels of Agreeableness showed stronger positive intervention effects on vividness ($d_{T2}$ = .28; $d_{T3}$ = .42; $d_{T4}$ = -.09; $d_{Overall}$ = .57 95%CI $d_{Overall}$ = .27; .88) than those with low levels of Agreeableness ($d_{T2}$ = .61; $d_{T3}$ = .05; $d_{T4}$ = -.32; $d_{Overall}$ = .33 95%CI $d_{Overall}$ = .03; .63; Fig 4).

## Discussion

To provide a systematic understanding of differential responses by personality traits in intervention contexts, we proposed a theoretical framework in which interventions are conceptualized as situations and described in terms of situational affordances. We conducted an initial test of this framework empirically using data from an evaluation study examining the FutureU app intervention which aims to strengthen future self-identification in order to increase future-oriented thinking and behavior. Our results showed that the intervention slightly increased vividness of the future self and that Conscientiousness and Extraversion were positively related to all three aspects of future self-identification, i.e., vividness of, valence towards, and relatedness to the future self. Based on the situational affordances provided by the intervention, we hypothesized that individuals with high levels of Conscientiousness, Openness to Experience, and/or Extraversion would benefit more from it by showing stronger increases in future self-identification. Contrary to our expectations, however, neither Conscientiousness nor Openness to Experience moderated intervention effects. And surprisingly, individuals

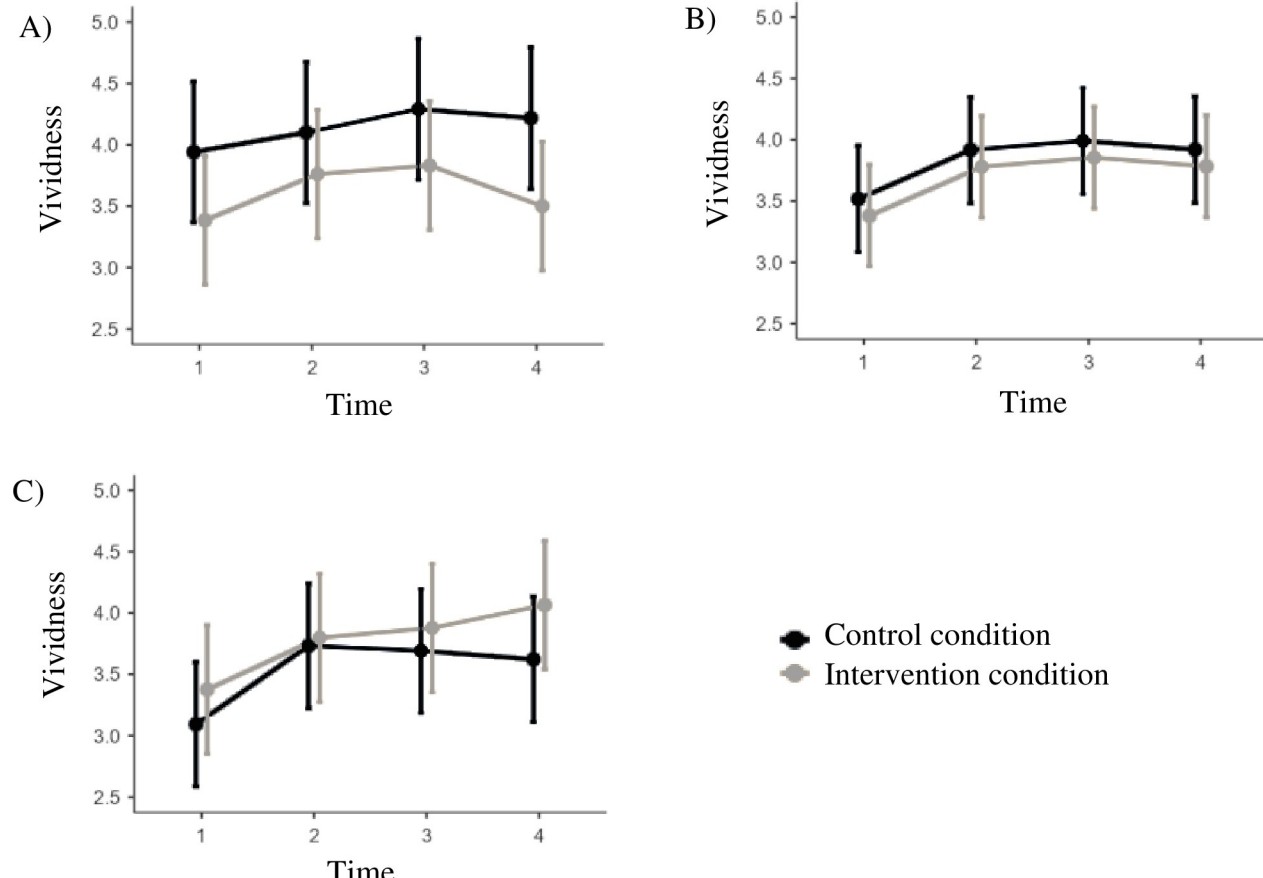

**Fig 4. Moderation of intervention effects on vividness by Agreeableness in the sensitivity analyses with plots of the intervention effects for participants.** A) low levels (M -1SD), B) average levels (M +/-1SD), and C) high levels (M +1SD) of Agreeableness. The range of the y-axis is truncated.

with low–rather than high–levels of Extraversion benefitted more from the intervention in terms of reporting higher future-self-identification. This moderation effect was particularly pronounced in the first week of the intervention where participants met their future self for the first time and thought about what kind of person this future self is.

The main effects of our analyses indicated that the FutureU intervention may be a promising intervention for stimulating future self-identification. However, although the small positive effect on vividness of the future self indicates the potential of the intervention, no intervention effects were apparent on valence towards and relatedness to the future self. Possibly, seeing a rendered version of one's future self is enough to increase vividness but may not affect feelings or perceived connection with it despite moments of interaction. Hence, the findings highlight the importance to further optimize the intervention and, at the same time, demonstrate the need to study intervention effects on the individual level. Given that heterogeneity among individuals is particularly pronounced in universal interventions (i.e., targeting an undifferentiated population, instead of individuals at risk or experiencing difficulties), it is important to further examine intervention effects among subgroups of individuals [25]; what may not work for everybody (i.e., main effects), could be effective for certain subgroups (i.e., moderation effects).

Regarding personality traits, the main effects suggest that personality traits may play an important role in the development of future self-identification in general. More specifically,

our results showed that Conscientiousness and Extraversion were both consistently related to all three aspects of future self-identification. Within the literature on future self-perspectives only limited attention has been paid to the role of personality. Hence, future research could further investigate the role of these two traits in the development of future self-identification to shed light on its underlying developmental processes.

Given that we found no support for our moderation hypotheses derived from the proposed affordance-based framework, do we have to abandon the framework altogether? In our view, this conclusion would be premature. For one, the lack of moderation concerning Conscientiousness may be explained by the nature of the control condition, which–upon closer inspection–arguably also afforded the expression of Conscientiousness. That is, in both the intervention and the control condition, individuals set goals, thereby creating situations specifically asking for planful and goal-directed behaviors, which are characteristic of Conscientiousness [16]. Although FutureU involved multiple situations that afforded behaviors related to Conscientiousness, we suspect that the increased opportunities to show these behaviors in the intervention condition were too weak to provide an additional stimulation in future self-identification.

Openness to Experience did not interact with intervention situations either. We expected that learning new skills, increasing self-insight, and (the virtual rendering of) the future self would afford the expression of Openness to Experience. Again, new skills and increased self-insight may have also played a role in the control condition, as goal-setting could have been a new skill for students and forced them to think about what they want to achieve, which, in turn, can increase self-insight. In that case, both conditions provided situations for the expression of Openness to Experience. Regarding the future self, the idea of interacting with the future self-avatar is perhaps less unconventional than we anticipated. Nowadays, interactions between people are often mediated by technologies, such as smartphones and computers, in which some form of avatar–symbolizing the people involved in the interaction (e.g., screen names, graphical icons, animated 3D-characters)–is used [26]. While we hypothesized that the FutureU-specific situations characterized by the virtual rendering of and interaction with the future self afforded the expression of curiosity, novel experiences, and unconventional ideas–all related to high levels of Openness to Experience–this may not be the case. Thus, FutureU intervention situations might not have specifically evoked the expression of Openness to Experience.

The finding that introverted individuals benefitted more from the intervention than more extraverted individuals may, in turn, be explained based on the app-based implementation. Although situations characterized by social interactions afford the expression of Extraversion, affordances may change when these social situations take place in a virtual, online context, such as an app. In fact, online social situations with the purpose to form new social relationships appear to afford the expression of low rather than high levels of Extraversion. Whereas introverted individuals are typically shy, withdrawn, and quiet in face-to-face interactions [16] this discomfort seems to diminish in online interactions in which communication is often text-based without live visual cues, and people can rewrite their responses and communicate at their own pace [27]. These situational characteristics enable more introverted individuals to express their true self by facilitating self-disclosure and feelings of intimacy [27, 28], which in turn, helped them to engage in the intervention situation and learn from it. This might explain why less extraverted individuals benefitted most from the app intervention and were indeed able to form a (new) social relationship with their future self. Note, however, that these explanations are based on a post-hoc application of the situational affordances framework and should thus be interpreted with caution.

In addition to our three focal traits, we explored the potentially interaction effects concerning Honesty-Humility, Emotionality, and Agreeableness. Except for Agreeableness, no evidence for interaction of these traits with intervention situations on the outcomes was found. Agreeableness interacted with intervention situations resulting in differential intervention effects on vividness of the future self, with individuals high on Agreeableness showing stronger intervention effect than individuals low on Agreeableness. However, this finding only occurred when excluding participants who experienced problems with the app and who failed the instructed attention checks embedded in the survey. High levels of Agreeableness capture sympathy, gentleness, and sentimentality [16]. Correspondingly, individuals high on Agreeableness are able to build and maintain positive social relationships [13]. FutureU situations characterized by meeting the future self and building a relationship with the digital avatar may thus afford the expression of behaviors related to high Agreeableness. However, given the surprising nature of this finding and the fact that it only emerged for a subsample of participants, it needs to be replicated before drawing conclusions.

## Limitations

The results of the current study should be considered in light of several limitations. First, FutureU consists of multiple situations which together form the intervention program. In the current study, the complete program was implemented, meaning that multiple intervention situations afforded the expression of the same trait and the combination of situations afforded the expression of multiple traits. As a consequence, it was impossible to test specific trait activation propositions in isolation, that is, which exact situation afforded the expression of which particular trait. Additionally, the affordances provided by situations to express certain personality traits is based on theory. However, how a situation is perceived could differ between individuals and may, therefore, also afford the expression of different personality traits for different people. For future research, it would be interesting to study situations and afforded expressions of traits in isolation as well as whether people perceive affordances of situations for the expressions of traits in the same way and in line with theoretical considerations.

Second, while an active control condition strengthens the research design for evaluating the effectiveness of an intervention, it was a limitation in the current study. Due to the goal-setting approach in the control condition, the two conditions shared some similarities, thereby arguably affording the expression of the same traits to some degree. This may have affected the ability to detect moderation effects of personality traits. Future research applying this framework in an intervention context should be aware of, and ideally consciously manipulate, the situations individuals in another condition encounter as well.

Third, although the sample size is quite large for an RCT examining an app intervention requiring personal contact, it can be considered relatively small for research studying personality. As a consequence, significant effects are less likely to show and effect size estimates are less precise. Therefore, more research into the description of interventions in terms of situations in order to explain differential treatment response based on personality traits is needed.

## Conclusion

We described and conducted a first test of a situational affordances framework to explain differential intervention response based on personality traits. This framework allowed us to retrospectively explain our results. Although the a priori predictions we made based on this framework were not confirmed, rejecting its usability in an intervention context based on the current study would be premature. The proposed framework showed potential for systematically understanding (differences in) intervention effects through intervention situations

affording the expression of certain personality traits. Our empirical test of the framework should be considered as a first step in the examination of its usability and requires more testing, for instance on different types of interventions. Moreover, the data we used provided us with the opportunity to examine differences in intervention response over time, but may not have had the optimal design to judge the application of the proposed framework. Before conclusions either confirming or rejecting situational affordances as a framework in an intervention context can be drawn, future research specifically designed for this aim should further test this. This could, for instance, be achieved by comparing a single-component intervention with a control condition without intervention situations. The framework of situational affordances can be useful to guide future research and strengthen the theoretical environment of current intervention research examining differential intervention response based on personality traits.

## Supporting information

**S1 File.**
(DOCX)

## Acknowledgments

The authors would like to thank Orb Amsterdam (www.orbamsterdam.com) for developing the smartphone application and for providing helpful suggestions.

## Declarations

**Preregistration:** The current study was pre-registered specifying the hypotheses and analysis plan (AsPredicted: https://aspredicted.org/95F_CDR [ANONYMIZED LINK]).

## Author Contributions

**Conceptualization:** Esther C. A. Mertens, Isabel Thielmann, Annalaura Nocentini, Jean-Louis van Gelder.

**Data curation:** Esther C. A. Mertens, Aniek M. Siezenga.

**Formal analysis:** Esther C. A. Mertens, Annalaura Nocentini.

**Funding acquisition:** Jean-Louis van Gelder.

**Investigation:** Aniek M. Siezenga.

**Methodology:** Esther C. A. Mertens, Jean-Louis van Gelder.

**Project administration:** Esther C. A. Mertens.

**Supervision:** Jean-Louis van Gelder.

**Writing – original draft:** Esther C. A. Mertens.

**Writing – review & editing:** Isabel Thielmann, Annalaura Nocentini, Aniek M. Siezenga, Jean-Louis van Gelder.

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
