## [Decision Letter · Decision Letter 0]

24 Jun 2024

PONE-D-24-00924Putting situational affordances in an intervention context: Personality as a moderator of intervention effectsPLOS ONE

Dear Dr. van Gelder,

Thank you for submitting your manuscript to PLOS ONE. After careful consideration, we feel that it has merit but does not fully meet PLOS ONE’s publication criteria as it currently stands. Therefore, we invite you to submit a revised version of the manuscript that addresses the points raised during the review process.

We look forward to receiving your revised manuscript.

Kind regards,

Frantisek Sudzina

Academic Editor

PLOS ONE

“This work was financially supported by a European Research Council Consolidator Grant (no. 772911-CRIMETIME).”

5. We note that Figure 2 in your submission contain copyrighted images. All PLOS content is published under the Creative Commons Attribution License (CC BY 4.0), which means that the manuscript, images, and Supporting Information files will be freely available online, and any third party is permitted to access, download, copy, distribute, and use these materials in any way, even commercially, with proper attribution. For more information, see our copyright guidelines: http://journals.plos.org/plosone/s/licenses-and-copyright.

1. You may seek permission from the original copyright holder of Figure(s) [#] to publish the content specifically under the CC BY 4.0 license.

Reviewers' comments:

Reviewer's Responses to Questions

**Comments to the Author**

1. Is the manuscript technically sound, and do the data support the conclusions?

Reviewer #1: Yes

Reviewer #2: Partly

2. Has the statistical analysis been performed appropriately and rigorously? 

Reviewer #1: Yes

Reviewer #2: Yes

3. Have the authors made all data underlying the findings in their manuscript fully available?

Reviewer #1: Yes

Reviewer #2: No

4. Is the manuscript presented in an intelligible fashion and written in standard English?

Reviewer #1: Yes

Reviewer #2: Yes

5. Review Comments to the Author

Reviewer #1: Review of PONE-24-00924: “Putting situational affordances in an intervention context: Personality as a moderator of intervention effects”

Many thanks for the opportunity to review this manuscript, which contains an interesting dataset based on state-of-the-art technology. Overall, the paper is well-written, and the analyses are performed competently. However, there are some problems with the theoretical and design-technical parts of the part, which I’ll explain below.

1. The introduction focuses on the effects of interventions on the expression of individual differences, i.e., that – depending on the intervention – some people may show stronger intervention-related outcomes based on their personality. However, the hypotheses, noted in the section on ‘The present study’ are about the ‘moderating role’ of personality on the intervention – outcome relation, which is something different. That is, the situational affordances framework argues that situations may moderate personality – criteria relations; here it is argued that personality moderates the situation – criteria relations. I know that – technically – this may entail the same analysis, but conceptually it is a different thing. Please make sure that you remain consistent through the manuscript which conceptualization is used.

2. Related to the above, it is unclear whether direct effects of personality and/or the situation were expected. Based on the absence of these direct effects in the preregistration, one would almost assume that no direct effects are hypothesized, but it would be highly unusual to expect pure moderator effects, so some direct effects are to be expected. There is some indication that the researchers expected a direct effect of the intervention in that FutureU ‘aims to strengthen people’s identification with their ‘future self’ in order to increase their future-oriented thinking and behavior’. Maybe it would be good to clearly state what the direct effects are in ‘The present study’ (although the authors might make a footnote in which they note that this wasn’t preregistered). Even less clear in the introduction is whether personality was expected to have a direct effect (which is actually the case in this study). It would be good to also make clear to the reader whether such direct effects were to be expected.

3. Although I appreciate the broad introduction on interventions and affordances, the sudden content shift to the ‘first test of this affordance-based intervention framework’ is large and is hard to follow for an unacquainted reader. I would either like to see – at the very beginning – a starting paragraph which introduces the central question and variables of this study before introducing the theoretical framework, or I would like the authors to make a better connection between the first (intervention and affordances) part and the second (FutureU) part. I prefer the first option. It also would be good to use some headers to help the reader navigate through the introduction because it contains a lot of different information.

4. Table 1 helps, but I would make sure that it is ordered in terms of the personality traits. Note that the outcome variable of interest is future self-identification; the authors might like to feature this in the table (or its heading) as well.

5. What I miss in the introduction, in Table 1, and in the methods section is how the FutureU intervention differs from the control condition in terms of the situational affordances. This is the most important variable of this study, and it is not clear from the introduction and beyond how the intervention is supposed to afford the expression of these three traits more than the control condition. Please explain. The control condition – or at least the kinds of situations that the control condition offers – might be featured in the Table 1 as well.

6. Interventions are usually targeted at behaviors, skills, and/or attitudes, and I understand from the introduction that the ultimate goal is to help participants to better set – and adhere – to goals. I also gather from the introduction (although it is not clearly stated as such) that future self-identification is thought to act as a mediator or moderator of the relation between the intervention and the ultimate goal of the intervention (i.e., more (future) goal-directed behaviors). To the reader, it is not clear how strong the evidence is that future self-identification indeed helps goal-directed behaviors and in what way (i.e., as a mediator or moderator). What kind of evidence can be brought to support the authors line of reasoning and how strong is this evidence (i.e., how ‘necessary’ is the variable future self-identification for the ultimate criterion)? Why didn’t the authors directly investigate the effect of the (interaction between the) intervention and personality on whether the participants set (more future-oriented/better) goals and/or whether they adhered to these goals?

7. Interaction effects – that are usually small – are notably difficult to find and replicate, so I wondered about the statistical power of the analyses, especially of the three-way interaction. Can the authors provide a power analysis?

8. As a reader, I prefer to first see a table with all correlations and descriptives (including M, SD, and reliabilities), containing (at least) gender, age, condition, personality variables, and (T1 through T4) criteria variables. In the supplement, this same table is preferably split in two (one for control condition, one for experimental condition). I especially wondered about the relations between the three criteria variables. What does factor-analysis show? Are these indeed independent factors, or might it be useful (and more parsimonious) to conduct the analyses on a higher-order (general) factor?

9. The effect of the intervention on vividness from T1 to T2 seems especially due to the high scores on vividness in T1 in the control group, so it is questionable whether this is actually due to the intervention, because there may be a ceiling effect for the level of vividness that participants experience. The sentence in the discussion that ‘The main effects … indicated that … FutureU … may be a promising intervention for stimulating future self-identification’ is too strong given the (lack of) evidence.

10. In general, my suggestion is to rewrite the manuscript by focusing on the intervention as a moderator of the personality – future self-identification relation (instead of personality as a moderator). This may help putting more in the foreground the important finding that Conscientiousness and Extraversion are (strong) personality predictors of future self-identification, something that – as far as I know – has not been studied before. I realize that this doesn’t strictly align with the preregistration, but I don’t see a problem with clarifying this in the manuscript (e.g., using a footnote). Good luck with the manuscript!

Reviewer #2: Thank you for the opportunity to review the manuscript (PONE-D-24-00924), entitled "Putting situational affordances in an intervention context: Personality as a moderator of

intervention effects." In their article, the authors suggest a theoretical framework for conceptualizing interventions as situations affording the expression of personality traits. The article has notable strengths, like the study design (RCT) and the empirical test of said framework. However, some issues could be addressed. Please find my comments below, I hope they help improving the article.

1. My first remark refers to the interpretation of results. In the preliminary results, the authors found an intervention effect for vividness but not for valence or relatedness. However, this finding was not discussed (apart from further optimizing the intervention). How could the intervention be optimized? Could one assume that people simply have a clearer image of themselves (i.e., more vividness) due to the presentation of their older avatar? If so, which adjustments would be possible to also address valence and relatedness?

2. The primary results revealed that there was only one significant interaction (for Extraversion), in contrast to their assumptions based on the framework. In the discussion, the authors mentioned that the lack of moderation could be explained by the control condition, which might have afforded certain traits as well. How could the intervention be modified so that the control condition does not afford the same trait? What would be the next steps?

3. Was a correction for multiple testing applied? If not, could the authors explain the rationale behind it?

4. Furthermore, I was thinking about alternative explanations for the results. Situational affordances are categorized as environmental variables that can clearly influence behavior, but person variables (like need or goals) also play an important role. Maybe it is not only about situational affordances but more about person situation fit (cf. Rauthmann, 2021)? And what about individual situation perception? Shouldn’t situation characteristics be included in the future to better understand whether the intervention actually affords the expression of certain traits? I think this could be discussed.

5. The authors state that open data were available and provide a link to an online platform, OSF. However, data were not openly available. The link to the data and code didn’t work. I guess this was due to a typo (“il” in the link instead of “io”, 295): However, even changing this typo does not solve the issue. Please add a View-only link in a potential revision of the article.

6. Concerning the figures, I was wondered if it would be possible to display the three groups (six trajectories) into one panel/graph? This way it would be easier to directly recognize the effects and compare them across groups. I would also recommend using confidence intervals around the point estimates (if possible). Finally, I would suggest mentioning the range of the y-axis in the note of the figure as it is truncated.

7. Generally, I think the limitations section could be more specific. The authors said, for instance, that the sample size was rather small for personality research (469). What would be a possible consequence of this? Why should it be a much larger sample for personality, and is the effect likely overestimated or underestimated?

Additional comments:

- The order of authors differs on the manuscript and on the completed form. Perhaps the authors would like to check this again.

- For vividness, the authors mentioned that items were answered on a 7-point Likert scale (228). Please consider this as a suggestion (I’m aware that many people use this wording), but I would recommend writing rating scale instead, as technically a Likert scale is a combination of items (cf. Likert, 1932).

References:

Likert, R. (1932). A technique for the measurement of attitudes. Archives of Psychology, 22 140, 55.

Rauthmann, J. F. (2021). Capturing interactions, correlations, fits, and transactions: A Person-Environment Relations Model. In The Handbook of Personality Dynamics and Processes (pp. 427–522). Elsevier. https://doi.org/10.1016/B978-0-12-813995-0.00018-2

6. PLOS authors have the option to publish the peer review history of their article (what does this mean?). If published, this will include your full peer review and any attached files.

Reviewer #1: No

Reviewer #2: No

---

## [Author Response · Author response to Decision Letter 0]

31 Jul 2024

Amsterdam 20 July 2024

Subject: Manuscript submission

Dear Dr. Chenette,

Enclosed please find our revised manuscript entitled “Putting situational affordances in an intervention context: How the interaction between personality and intervention situations can help us explain differential intervention response” (PONE-D-24-00924; previously titled “Putting situational affordances in an intervention context: Personality as a moderator of intervention effects”). We would like to thank the Reviewers for their kind words and thoughtful comments. We believe the revisions have greatly improved the manuscript. In the letter, we explain how we dealt with the main concerns raised by the reviewers. Point by point responses are provided below the letter. 

In line with the suggestion of Reviewer 1, we have thoroughly checked the manuscript regarding the correct formulation of the moderation effect in line with the situational affordances framework. Furthermore, we made sure that the materials and data of our project are publicly available via the OSF repository. We have also amended the Role of Funder statement as suggested: “This work was financially supported by a European Research Council Consolidator Grant (no. 772911-CRIMETIME). The funder had no role in study design, data collection and analysis, decision to publish, or preparation of the manuscript.”

Regarding Figure 2, presenting the screenshots of the FutureU smartphone application, I – the corresponding author – am the copyright holder of this application. The first author and I have completed the Content Permission Form and explicitly mention in the Figure that we have consent from the copyright holder. In order to facilitate review and evaluation of the revisions, the changes in the manuscript are highlighted and our responses to the Reviewers’ comments are listed in the Appendix.

Many thanks again for considering our work. We look forward to hear back from you.

Sincerely,

On behalf of my co-authors Mertens, Thielmann, Nocentini, and Siezenga,

Jean-Louis van Gelder

j.vangelder@csl.mpg.de

Appendix Response to Reviewers

Reviewer #1: 

1. The introduction focuses on the effects of interventions on the expression of individual differences, i.e., that – depending on the intervention – some people may show stronger intervention-related outcomes based on their personality. However, the hypotheses, noted in the section on ‘The present study’ are about the ‘moderating role’ of personality on the intervention – outcome relation, which is something different. That is, the situational affordances framework argues that situations may moderate personality – criteria relations; here it is argued that personality moderates the situation – criteria relations. I know that – technically – this may entail the same analysis, but conceptually it is a different thing. Please make sure that you remain consistent through the manuscript which conceptualization is used.

Response: We agree and thank the Reviewer for pointing this out. We have adjusted the title and text accordingly throughout the manuscript. 

For example, “The present study aimed to illuminate the potential of situational affordances for improving our understanding of personality influences in intervention contexts. To this end, we described the FutureU intervention in terms of intervention situations and corresponding affordances and examined whether the intervention situation (compared to the control condition) moderated the relation between personality traits and intervention effects.” (p. 8)

2. Related to the above, it is unclear whether direct effects of personality and/or the situation were expected. Based on the absence of these direct effects in the preregistration, one would almost assume that no direct effects are hypothesized, but it would be highly unusual to expect pure moderator effects, so some direct effects are to be expected. There is some indication that the researchers expected a direct effect of the intervention in that FutureU ‘aims to strengthen people’s identification with their ‘future self’ in order to increase their future-oriented thinking and behavior’. Maybe it would be good to clearly state what the direct effects are in ‘The present study’ (although the authors might make a footnote in which they note that this wasn’t preregistered). Even less clear in the introduction is whether personality was expected to have a direct effect (which is actually the case in this study). It would be good to also make clear to the reader whether such direct effects were to be expected.

Response: Following the recommendation of the Reviewer we now explicitly mention the direct effects in ‘The present study’. 

“Furthermore, we expected a direct, positive effect of the FutureU intervention on future self-identification given that it aims to increase people’s identification with their future self. We had no specific hypotheses regarding direct effects of personality traits on future self-identification.”(p. 9)

3. Although I appreciate the broad introduction on interventions and affordances, the sudden content shift to the ‘first test of this affordance-based intervention framework’ is large and is hard to follow for an unacquainted reader. I would either like to see – at the very beginning – a starting paragraph which introduces the central question and variables of this study before introducing the theoretical framework, or I would like the authors to make a better connection between the first (intervention and affordances) part and the second (FutureU) part. I prefer the first option. It also would be good to use some headers to help the reader navigate through the introduction because it contains a lot of different information.

Response: This was a helpful and clarifying comment. We have now added the central aim and the specific variables of the present study in the first paragraph in the introduction. Additionally, we added headers in the introduction to help the reader navigate it.

“In the present study, we propose that interventions place individuals in different situations that may afford the expression of different personality traits and, thereby, allow different individuals to benefit more, or less, from that specific intervention. More specifically, we examined whether intervention situations created by the FutureU intervention moderated the relation between individuals’ personality traits and the intervention’s effects on future self-identification.” (p. 3)

4. Table 1 helps, but I would make sure that it is ordered in terms of the personality traits. Note that the outcome variable of interest is future self-identification; the authors might like to feature this in the table (or its heading) as well.

Response: We have reordered Table 1 as suggested. As this is a representation of the information provided in the text, we put the focus here on which intervention situations could afford the expression of certain personality traits rather than the outcomes of the intervention itself.

5. What I miss in the introduction, in Table 1, and in the methods section is how the FutureU intervention differs from the control condition in terms of the situational affordances. This is the most important variable of this study, and it is not clear from the introduction and beyond how the intervention is supposed to afford the expression of these three traits more than the control condition. Please explain. The control condition – or at least the kinds of situations that the control condition offers – might be featured in the Table 1 as well.

Response: Participants in the intervention condition set goals and received the FutureU intervention, while the control condition only set goals. In terms of intervention situations, individuals in the intervention condition encountered all the intervention situations mentioned in Table 1, whereas individuals in the control condition only encountered an intervention situation affording goal-directed behavior. We now explicitly describe this difference in the introduction and added a note below Table 1 stating that participants in the control condition did not encounter the described intervention situations except for setting goals. Additionally, we now emphasize it again in the method section.

For example, “The FutureU intervention situations were not apparent in the control condition, except for setting goals. As in the intervention condition, people in the control condition set goals, creating a situation affording goal-directed behaviors, but in contrast to the intervention condition they received no further intervention and, thus, did not encounter other intervention situations.” (p. 8)

“Hence, besides setting weekly goals for three weeks, the situations encountered by the participants in their daily lives were not affected by an intervention.” (p. 12)

6. Interventions are usually targeted at behaviors, skills, and/or attitudes, and I understand from the introduction that the ultimate goal is to help participants to better set – and adhere – to goals. I also gather from the introduction (although it is not clearly stated as such) that future self-identification is thought to act as a mediator or moderator of the relation between the intervention and the ultimate goal of the intervention (i.e., more (future) goal-directed behaviors). To the reader, it is not clear how strong the evidence is that future self-identification indeed helps goal-directed behaviors and in what way (i.e., as a mediator or moderator). What kind of evidence can be brought to support the authors line of reasoning and how strong is this evidence (i.e., how ‘necessary’ is the variable future self-identification for the ultimate criterion)? Why didn’t the authors directly investigate the effect of the (interaction between the) intervention and personality on whether the participants set (more future-oriented/better) goals and/or whether they adhered to these goals?

Response: We agree with the Reviewer that these are all interesting questions. However, the aim of the study was to examine the applicability of the situational affordances framework in an intervention context. Therefore, we focused on moderation effects of situations on the expression of personality. We used the proximal outcomes of the FutureU intervention, i.e., future self-identification variables, rather than distal outcomes. Generally, intervention effects are more pronounced immediately after the intervention in the proximal outcomes than in the distal outcomes. Hence, if there would be an interaction effect with personality traits, these would most likely show on the variables representing the proximal outcome of future self-identification. Questions addressing the effectiveness of the intervention and testing of the theoretical intervention model are important, though beyond the scope of this paper.

7. Interaction effects – that are usually small – are notably difficult to find and replicate, so I wondered about the statistical power of the analyses, especially of the three-way interaction. Can the authors provide a power analysis?

Response: We agree with the Reviewer that interaction effects are usually small. We ran a power analysis using G*Power. With an effect size of f = 0.143 and α error probability of 0.05 the power (1 – β error probability) to detect a three-way interaction is 0.95. We have added this power analysis as a footnote in the analyses section.

“Power analysis ran in G*Power showed a power of .95 to detect a three-way interaction, with an f = .14 and an α error probability of 0.05.” (p.14)

8. As a reader, I prefer to first see a table with all correlations and descriptives (including M, SD, and reliabilities), containing (at least) gender, age, condition, personality variables, and (T1 through T4) criteria variables. In the supplement, this same table is preferably split in two (one for control condition, one for experimental condition). I especially wondered about the relations between the three criteria variables. What does factor-analysis show? Are these indeed independent factors, or might it be useful (and more parsimonious) to conduct the analyses on a higher-order (general) factor?

Response: We have added a Table with the correlations and descriptives of personality and the outcome variables for the complete sample and moved the Table with descriptives per condition to the supporting materials. The descriptives of gender, age and condition are reported for the total sample in the method section under the heading “Participants”. The reliabilities of the separate instruments are also reported in the method section under the heading “Measurements”.

Regarding the criteria variables, factor analyses suggest a two-factor structure at each timepoint representing the subscales vividness and relatedness. The item representing valence loads on the factor of vividness. It may be possible to have a hierarchical model where the items load on two latent factors representing vividness and relatedness that, in turn, load on one latent factor representing future self-identification. However, this hierarchical model would add parameters to be estimated, making the models more complex rather than more parsimonious. 

9. The effect of the intervention on vividness from T1 to T2 seems especially due to the high scores on vividness in T1 in the control group, so it is questionable whether this is actually due to the intervention, because there may be a ceiling effect for the level of vividness that participants experience. The sentence in the discussion that ‘The main effects … indicated that … FutureU … may be a promising intervention for stimulating future self-identification’ is too strong given the (lack of) evidence.

Response: We do not agree with the Reviewer that there may be a ceiling effect for the level of vividness for participants in the control condition who scored higher on vividness at T1 than participants in the intervention condition. Vividness is measured on a 7-point scale. At T1, participants in the control condition have an average score of 3.50, indicating that they can increase another 3.50 points. In addition, the calculated effect sizes concern the change in scores (see formula p. 14), that is, the change per condition participants show, e.g., from T1 to T2. Differences at T1 between conditions do not affect this effect size calculation.

We do agree with the Reviewer though that we cannot draw firm conclusions about the effectiveness of the FutureU intervention and therefore mention that the intervention “may be promising”, that there was a “small positive effect”, explicitly emphasizing that there were no effects on valence and relatedness, and stress “the importance to further optimize the intervention” (p. 22).

10. In general, my suggestion is to rewrite the manuscript by focusing on the intervention as a moderator of the personality – future self-identification relation (instead of personality as a moderator). This may help putting more in the foreground the important finding that Conscientiousness and Extraversion are (strong) personality predictors of future self-identification, something that – as far as I know – has not been studied before. I realize that this doesn’t strictly align with the preregistration, but I don’t see a problem with clarifying this in the manuscript (e.g., using a footnote). 

Response: We thank the Reviewer for the suggestion to focus the manuscript on the intervention as a moderator. We have adjusted the manuscript (including the title) to emphasize that we examined intervention situations as a moderator of the relation between personality traits and future self-identification. We now also explicitly mention in ‘The present study’ that we included the direct effects of the intervention and personality traits on future self-identification. However, we do not address these direct effects in depth.

For example, “To this end, we described the FutureU intervention in terms of intervention situations and corresponding affordances and examined whether the intervention situation (compared to the control condition) moderated the relation between personality traits and intervention effects.” (p. 8)

“Interactions of personality trait

---

## [Editor Report · Decision Letter 1]

7 Aug 2024

Putting situational affordances in an intervention context: How the interaction between personality and intervention situations can help us explain differential intervention responses

PONE-D-24-00924R1

Dear Dr. van Gelder,

We’re pleased to inform you that your manuscript has been judged scientifically suitable for publication and will be formally accepted for publication once it meets all outstanding technical requirements.

Kind regards,

Frantisek Sudzina

Academic Editor

PLOS ONE
---

## [Editor Report · Acceptance letter]

9 Sep 2024

PONE-D-24-00924R1 

PLOS ONE

Dear Dr. van Gelder, 

I'm pleased to inform you that your manuscript has been deemed suitable for publication in PLOS ONE. Congratulations! Your manuscript is now being handed over to our production team.

Kind regards, 

on behalf of

Dr. Frantisek Sudzina 

Academic Editor

PLOS ONE